# GRAPH RECURRENT ATTENTION NETWORKS FOR SOLVING SATISFIABILITY PROBLEMS

## ABSTRACT

In recent years, the use of deep learning for solving Boolean Satisfiability (SAT) problems has gained significant interest. This paper advances such neural-based methods by introducing a **G**raph **r**ecurrent **a**ttention **n**etwork for **SAT** (GranSAT). GranSAT employs two innovative steps to guide the network to search towards satisfaction of clauses: (1) evaluating the truth degree of each clause based on t-conorm fuzzy logic operators, and (2) updating assignments with attention mechanisms, closely aligning with distributed local search methods. Logical states are coupled with recurrently updated hidden states that are used to compute attention values, allowing the model to refine fuzzy assignments while retaining information from previous updates. Experimental results on crafted and random SAT benchmarks demonstrate that GranSAT outperforms existing neural SAT solvers in both performance and generalization. Furthermore, when combined with local search post-processors, GranSAT achieves state-of-the-art performance on random instances, showcasing its effectiveness in solving SAT problems.

## 1 INTRODUCTION

In recent years, there has been growing interest in using continuous methods for solving Boolean Satisfiability (SAT), a fundamental problem in computer science proven to be NP-complete (Cook, 1971). The earliest approaches involved the use of deep learning, attempting to predict the satisfiability of a given problem from one-bit supervision (Selsam et al., 2019). On the other hand, some works have focused on using continuous optimization techniques such as Fourier analysis of Boolean functions (Kyrillidis et al., 2020) and gradient descent with matrix representations (Sato & Kojima, 2019; 2021), to find satisfying assignments for given propositional formulas. Given that the computation of satisfiable assignments is a core problem in many AI, engineering, and scientific applications (Biere et al., 2009), we specifically aim to establish a novel deep learning based method for finding satisfying assignments of a given SAT instance.

Graph Neural Networks (GNNs) have been considered as a promising building block for solving combinatorial optimization problems (Cappart et al., 2023), as well as for bridging the gap between neural networks and symbolic reasoning (Lamb et al., 2020). In SAT, a wide variety of GNN-based solvers have been proposed, with objectives ranging from satisfiability prediction (Selsam et al., 2019) to assignment prediction (Chang & Liu, 2025). This paper advances such neural methods with **G**raph **r**ecurrent **a**ttention **n**etworks for **SAT** (GranSAT), a novel GNN that operates with novel attention mechanisms tailored to work with fuzzy logic variable assignments. GranSAT employs a two-step procedure: (1) evaluating each clause's satisfaction state from the fuzzy variable assignments using t-conorms, and (2) updating the variable assignments using attention mechanisms. This closely aligns with distributed local search, where the clause evaluation and variable assignment updates are done in a distributed manner. By keeping intermediate fuzzy assignments, GranSAT can learn to perform updates that are more in line with discrete local search methods.

To evaluate the effectiveness of our proposed model, we conduct extensive experiments on various SAT benchmarks, including both crafted and random instances. Our results demonstrate that GranSAT outperforms existing neural SAT solvers in terms of performance, while also showing strong generalization capabilities. Furthermore, to refine the fuzzy assignments predicted by GranSAT, we propose to use local search methods for post-processing. We adapt both discrete (Chu et al., 2023) and continuous (Sato & Kojima, 2019) local search methods to handle the continuous

outputs of GranSAT. From experiments on benchmark instances, we demonstrate that when combined with local search post-processors, GranSAT achieves state-of-the-art performance on random instances, showcasing its effectiveness in solving SAT problems.

Our main contributions are summarized as follows:

- We present GranSAT, an attention-based GNN model for SAT solving that operates on fuzzy logic variable assignments, closely aligning with distributed local search methods.
- We propose to use local search methods for post-processing to find satisfying assignments from the fuzzy assignments predicted by GranSAT.
- We conduct extensive experiments on various benchmarks, demonstrating the effectiveness of GranSAT compared to existing state-of-the-art neural and discrete SAT solvers.

## 2 RELATED WORK

### 2.1 NEURAL SAT SOLVERS

Over the years, multiple neural SAT solvers have been proposed, each with different architectures and objectives. These solvers are typically divided into two categories: end-to-end approaches, and hybrid approaches. End-to-end neural solvers aim to learn the entire solving process using neural networks, with the goal of either predicting the satisfiability of a given instance or generating satisfying assignments for them. Due to the simplicity of converting SAT instances into graph representations, Recurrent Neural Networks (RNNs) and Graph Neural Networks (GNNs) have been the standard in pioneering models such as NeuroSAT (Selsam et al., 2019), SAT-GATv2 (Chang & Liu, 2025), and DG-DAGRNN (Amizadeh et al., 2019). While they are not up to the standard of discrete solvers, these models have shown strong results regarding generalization to larger instances of the same family they have trained on.

On the other hand, hybrid approaches aim to leverage the strengths of both neural networks and modern SAT solvers by using neural predictions as heuristics for the solvers. These methods also often involve the use of RNNs or GNNs, which in this case is used to learn various heuristics such as variable states heuristics (Selsam & Bjørner, 2019; Wang et al., 2024) and branching heuristics (Kurin et al., 2020). The proposed hybrid solvers show competitive performance against modern SAT solvers, underscoring the potential of integrating neural networks in SAT solvers.

Our approach relates to both categories; while GranSAT is a neural solver that learns to solve SAT problems, it can be used with continuous or discrete local search algorithms to refine its output and find valid satisfying assignments. Furthermore, our architecture is unique in that it employs logical states which represent the fuzzy assignments of variables and satisfaction states of clauses, along with recurrent hidden states that store memories from previous updates, making the update process more in line with discrete methods.

### 2.2 LOCAL SEARCH SAT SOLVERS

While modern SAT solvers based on complete methods such as CDCL and DPLL are extremely strong, local search methods have the advantage of being able to solve large and randomly generated SAT instances efficiently. GSAT, one of the first Stochastic Local Search (SLS) solvers, uses an iterative improvement algorithm that starts with a random assignment and flips variable values to reduce the number of unsatisfied clauses (Selman & Kautz, 1993). More recently, Sparrow2Riss (Heule et al., 2018) won the random track of the SAT Competition 2018, demonstrating the effectiveness of SLS methods in solving large and randomly generated SAT instances.

More recently, major advancements in computational hardware such as GPUs have led to the development of multiple continuous solvers. MatSat (Sato & Kojima, 2019; 2021) is a continuous solver that uses matrix representations of SAT instances along with gradient descent, and demonstrates competitive performance against state-of-the-art local search solvers such as Sparrow2Riss. FourierSAT (Kyrillidis et al., 2020; Cen et al., 2025) is another continuous local search solver based on the Fourier analysis of Boolean functions, showing promising results on a variety of problems including SAT as well as MaxSAT.

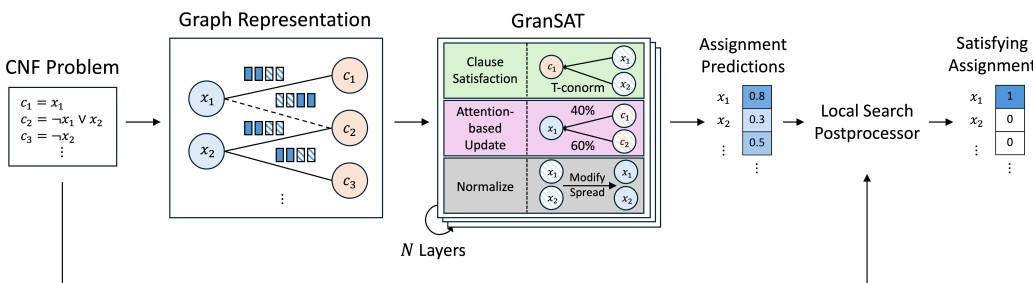

Figure 1: Overview of the proposed architecture for GranSAT with local search post-processing.

## 3 PRELIMINARIES

### 3.1 BOOLEAN SATISFIABILITY

The Boolean Satisfiability Problem (SAT) is the problem of determining whether there exists an assignment to a set of Boolean variables that makes a given propositional logic formula evaluate to true. The formula is typically written in Conjunctive Normal Form (CNF), which is a conjunction ($\wedge$) of clauses, where each clause is a disjunction ($\vee$) of literals, which are Boolean variables or its negations ($\neg$). Formally, given a CNF formula $\mathcal{F} = c_1 \wedge c_2 \wedge \cdots \wedge c_m$ over Boolean variables $\{x_1, x_2, \ldots, x_n\}$, a satisfying assignment is one where each clause $c_i$ contains at least one literal that evaluates to true.

### 3.2 GRAPH NEURAL NETWORKS

Graph Neural Networks (GNNs) are neural architectures designed for learning representations over graphs, and is often used in the SAT setting as CNF formulas are easily convertible to graphs (Section 4.1). GNNs mainly consist of message passing layers, where each node aggregates messages from its neighbors to update its own representation. Velickovic et al. (2018) introduced an attention mechanism to the message passing process, thus allowing each node to focus on more relevant neighbors. Given a node $i$ and its neighbor $j$, the attention coefficient $\alpha_{ij}$ is computed as

$$
\begin{aligned}
s_{ij} &= \text{LeakyReLU}\left(\mathbf{a}^\top [\mathbf{W}\mathbf{h}_i \| \mathbf{W}\mathbf{h}_j]\right) \\
\alpha_{ij} &= \frac{\exp(s_{ij})}{\sum_{k \in \mathcal{N}_i} \exp(s_{ik})},
\end{aligned}
\tag{1}
$$

where $\boldsymbol{h}_i$ and $\boldsymbol{h}_j$ are the feature vectors of nodes $i$ and $j$, $\mathbf{W}$ is a learnable linear transformation, $\mathbf{a}$ is a learnable weight vector. Furthermore, $\|$ denotes concatenation, and $\mathcal{N}_i$ is the set of neighbors of $i$. The updated node representation is then computed as a weighted sum of the transformed neighbor features

$$
\boldsymbol{h}_i' = \sigma\left(\sum_{j \in \mathcal{N}_i} \alpha_{ij} \mathbf{W}\mathbf{h}_j\right),
$$

where $\sigma$ is a non-linear activation function. The attention mechanism can additionally consist of multiple heads, where multiple sets of attention coefficients are computed in parallel and finally averaged, allowing the model to capture different aspects of the graph structure.

## 4 GRANSAT: GRAPH RECURRENT ATTENTION NETWORKS FOR SAT

This section introduces GranSAT, a novel GNN architecture equipped with attention mechanisms specifically designed to operate with fuzzy logic variable assignments. GranSAT operates mainly on a two-step procedure. First, it computes the satisfaction state of each clause from the assignments using t-conorms (Klement et al., 2000), a fuzzy logic operator for disjunctions. Then, it updates the assignments using attention mechanisms, with message passing formulations specifically designed to update the fuzzy assignments of the variables, in order to satisfy as many clauses as possible. The entire pipeline of GranSAT is illustrated in Figure 1.

Specifically, each variable $\boldsymbol{h}_i = (\boldsymbol{h}_i^L, \boldsymbol{h}_i^H)$ has a logical state $\boldsymbol{h}_i^L \in [0,1]^d$ ($d$ is the dimension of the hidden state) that represents the soft assignments of variables and satisfaction state of clauses, and a hidden state $\boldsymbol{h}_i^H \in \mathbb{R}^d$ that stores recurrent memory that gets continuously updated each layer. For readability, we interchangeably use $\boldsymbol{x}_i$ with $\boldsymbol{h}_i$ and $\boldsymbol{c}_j$ with $\boldsymbol{h}_{j+n}$.

## 4.1 Graph Representation

In prior works, multiple different graph representations have been proposed for SAT solving with GNNs. The most common representation is with a bipartite graph, with the nodes consisting of either the variables and clauses (Kurin et al., 2020), or literals and clauses (Selsam et al., 2019; Li et al., 2024). In our work, we focus on the former, with edge features specifically embedding the polarity of the literals.

Formally, each edge is associated with a multi-dimensional feature vector that represents the polarity as well as structural properties of each literal within the clause. Each clause $\boldsymbol{c}_j$ is associated with a literal set $\mathcal{N}_{j+n} = \{\mathcal{N}_{j+n}^+ \cup \mathcal{N}_{j+n}^-\}$, where $\mathcal{N}_{j+n}^+$ and $\mathcal{N}_{j+n}^-$ consists of positive and negative appearing literals, respectively. Formally, the edge feature vector $\boldsymbol{e}_{i,j+n}$ for a literal and its clause is defined as

$$\boldsymbol{e}_{i,j+n} = \begin{cases} \left( \frac{1}{|\mathcal{N}_{j+n}^+|}, \frac{1}{|\mathcal{N}_{j+n}^-|}, 0, 0 \right) & \text{if } i \in \mathcal{N}_{j+n}^+ \\ \left( 0, 0, \frac{1}{|\mathcal{N}_{j+n}^+|}, \frac{1}{|\mathcal{N}_{j+n}^-|} \right) & \text{if } i \in \mathcal{N}_{j+n}^- \end{cases},$$

where elements with a denominator of 0 are set to 0. These feature vectors are designed to capture the polarity and structural information necessary to determine the influence of each variable on the clause satisfaction during attention computation.

## 4.2 Clause Satisfaction

To compute the logical satisfaction state of each clause node, we use t-conorms, a fuzzy logic operator for disjunctions. Let $\boldsymbol{x}^L \in [0,1]^d$ be the current assignment vector. For each literal $\boldsymbol{l}_i$, we compute its soft truth value:

$$\boldsymbol{l}_i = \begin{cases} \boldsymbol{x}_i^L & \text{if } i \in \mathcal{N}_{j+n}^+ \\ 1 - \boldsymbol{x}_i^L & \text{if } i \in \mathcal{N}_{j+n}^- \end{cases}$$

We then compute the clause satisfaction $\boldsymbol{c}_j^L$ with one of the following t-conorms $T^\vee$:

- **Gödel:** $T^\vee(\boldsymbol{l}_{i_1}, \boldsymbol{l}_{i_2}, \dots, \boldsymbol{l}_{i_k}) = \max(\boldsymbol{l}_{i_1}, \boldsymbol{l}_{i_2}, \dots, \boldsymbol{l}_{i_k})$

- **Łukasiewicz:** $T^\vee(\{\boldsymbol{l}_i\}_{i \in \mathcal{N}_{j+n}}) = \min\left(1, \sum_{i_k \in \mathcal{N}_{j+n}} \boldsymbol{l}_i\right)$

- **Product:** $T^\vee(\boldsymbol{l}_{i_1}, \boldsymbol{l}_{i_2}, \dots, \boldsymbol{l}_{i_k}) = 1 - \prod_{j=1}^k (1 - \boldsymbol{l}_{i_j})$

The specific t-conorm $T^\vee$ is selected based on a user-defined configuration and applied across all literals in the clause.

## 4.3 Attention-Based Logical Updates

In this section, we describe the update procedure of the logical states of the variables using attention mechanisms. First, we compute the hidden states using Gated Recurrent Units (GRUs) (Chung et al., 2014), in order to retain information from previous updates. Then, we compute attention scores for each variable–clause pair using the hidden states, and finally update the logical states of the variables using the attention scores and current logical states.

**Hidden State Update.** To keep both the current state and global context of each variable, we use a dual embedding approach that combines logical states and hidden states. In SAT solving with GNNs, hidden state updates are typically updated using RNNs of some form, with the main method being LSTMs (Selsam et al., 2019; Chang et al., 2022). Building on this, our hidden states are updated using Gated Recurrent Units (GRUs), while logical states are updated using attention-based

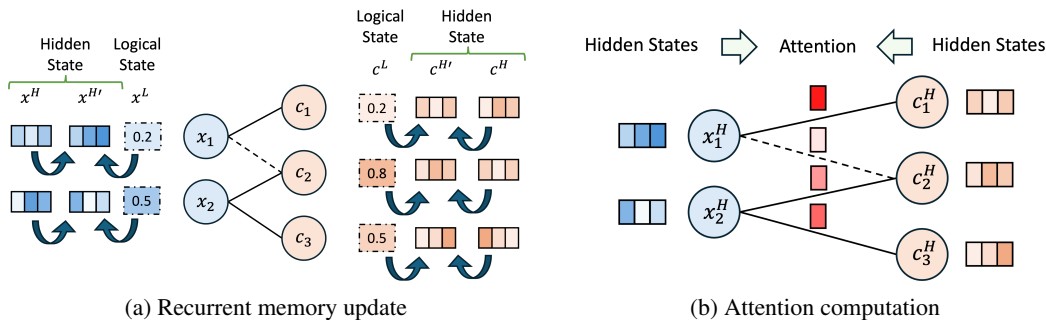

Figure 2: Attention based logical update

message passing. GRUs are chosen for their lower computational cost compared to LSTMs, while still maintaining the ability to capture long-term dependencies. The hidden state for each node $\boldsymbol{h}_i^H$ is updated using a GRU cell as follows (Figure 2a):

$$\begin{cases} \boldsymbol{x}_i^{H'} &= \text{GRUCell}_{\text{x}}\left(\boldsymbol{x}_i^L, \boldsymbol{x}_i^H\right) \\ \neg\boldsymbol{x}_i^{H'} &= \text{GRUCell}_{\text{x}}\left(1 - \boldsymbol{x}_i^L, \neg\boldsymbol{x}_i^H\right) \end{cases}, \quad \boldsymbol{c}_i^{H'} = \text{GRUCell}_{\text{c}}\left(1 - \boldsymbol{c}_i^L, \boldsymbol{c}_i^H\right),$$

Here, we prepare two pairs of hidden states for variables, one for positive literals, and one for negative literals. The positive hidden state $\boldsymbol{x}_i^H$ is updated using the current logical state $\boldsymbol{x}_i^L$, while the negative hidden state $\boldsymbol{c}_i^H$ is updated using the negated clause satisfaction $1 - \boldsymbol{c}_i^L$. Likewise, the clause hidden state $\boldsymbol{c}_i^H$ is updated using the negated clause satisfaction $1 - \boldsymbol{c}_i^L$, as clauses that are unsatisfied should have a higher influence on the variable updates. The recurrent nature of GRUs allows the model to retain information from previous updates, enabling it to make more informed decisions during the logic state update. A more detailed description of the hidden state update is provided in Appendix A.

**Attention Computation.** Using the hidden states, we compute the attention scores for each variable clause pair (Figure 2b). Let $\boldsymbol{x}_i$ and $\boldsymbol{c}_{j-n}$ be the current logical states of variable node $i$ and clause node $j-n$, respectively. For each edge $(i, j)$ with edge features $\boldsymbol{e}_{i,j}$, we compute the attention coefficients with the following equation:

$$s_{ij} = \text{LeakyReLU}\left(\mathbf{a}^\top \left[\mathbf{W}_l \boldsymbol{x}_i^H \| \mathbf{W}_r \boldsymbol{c}_{j-n}^H \| \mathbf{W}_e \boldsymbol{e}_{i,j}\right]\right)$$

Here, $\mathbf{W}_l$, $\mathbf{W}_r$, and $\mathbf{W}_e$ are learned linear transformations, and $\mathbf{a}$ is a learnable attention weight vector. Furthermore, $\boldsymbol{x}_i^H$ is replaced with $\neg\boldsymbol{x}_i^H$ if $i \in \mathcal{N}_{j+n}^-$, in order to reflect the polarity of the literal. The attention weights are then normalized across the neighborhood of each variable using Equation 1, to finally obtain the attention coefficients $\alpha_{ij}$.

**Variable Logical State Update.** Finally, the logical state of each variable $\boldsymbol{x}_i^L$ is updated using the attention coefficients and literal polarity $\pi_{ij} \in \{+1, -1\}$ as follows:

$$\boldsymbol{x}_i^{L'} = \sum_{j \in \mathcal{N}_{i+n}} \alpha_{ij}\left(\boldsymbol{x}_i^L + \pi_{ij}(1 - \boldsymbol{c}_{j-n}^L)\right)$$

This update rule effectively reflects the direction in which the variable should be adjusted depending on the unsatisfiability of the clause. If the clause is highly unsatisfied, the influence will be larger, and the variable will be updated towards satisfying the clause. The attention coefficients then determine which clauses to focus on, allowing the model to learn to prioritize certain clauses over others. Notably, as this update rule may result in values outside the $[0, 1]$ range, we clamp the values to ensure they remain within this range.

## 4.4 NORMALIZATION

To ensure that the updated logical states are not cluttered or spread out, we perform normalization with the normalized tunable sigmoid function (Dini, 2010). This function provides a smooth,

Figure 3: Variable logical state update via attention and current logical states.

bounded, and curvature-adjustable mapping from $[0,1]^d \to [0,1]^d$, making it ideal for controlling the spread of logical states. Using a learnable curvature parameter $\boldsymbol{k} \in [-1,1]^d$, the normalized output is

$$\boldsymbol{x}_i^{L'} = \frac{1}{2}\left(\frac{(1-\boldsymbol{k})(2\boldsymbol{x}_i^L - 1)}{1 + \boldsymbol{k} - 2\boldsymbol{k}\,|2\boldsymbol{x}_i^L - 1|} + 1\right)$$

This formulation allows us to control the spread around $\boldsymbol{x}_i^L = 0.5$, with a higher $\boldsymbol{k}$ value leading to a more cluttered distribution around the center. By integrating this normalization, we aim to achieve stable training by allowing the model to control the spread of logical states.

## 4.5 LOSS FUNCTION

To train the model in an unsupervised manner, we define a loss based on the final clause satisfactions. Let $\bar{c}_j \in [0,1]$ denote the scalar satisfaction of clause $j$ obtained by applying the chosen t-conorm to the final literal predictions (derived from the average over the dimensions of all variable logical states). We then minimize

$$\mathcal{L} = -\frac{1}{m}\sum_{j=1}^{m}\log \bar{c}_j. \tag{2}$$

This corresponds to a Binary Cross-Entropy objective with target 1 for satisfiable instances.

## 4.6 LOCAL SEARCH POST-PROCESSING

To search for satisfying assignments from the fuzzy assignments predicted by GranSAT, we propose to use local search methods for post-processing. We adapt both discrete and continuous local search methods to handle the continuous outputs of GranSAT.

**MatSat$^+$**  As the outputs of GranSAT are continuous, one approach is to use a continuous local search solver to refine the outputs. To this end, we use a customized version of MatSat (Sato & Kojima, 2019; 2021), which uses the predicted outputs as the initial assignment.

**NuWLS$^+$**  As a different approach, we attempt to use a discrete local search MaxSAT solver to refine the outputs of GranSAT. We give the solver a customized problem instance, where hard clauses (clauses that must be satisfied) are defined as the clauses in the original SAT instance, and soft clauses (clauses that can be violated) are unit clauses for every variable $x_i$, with weights and polarity defined using the predicted output $p_i$ as follows:

$$\begin{cases}(x_i), & \text{weight } w_i = p_i & \text{if } p_i \geq 0.5 \\ (\neg x_i), & \text{weight } w_i = 1 - p_i & \text{if } p_i < 0.5\end{cases}$$

This formulation uses the magnitude and sign of the predicted assignment to set the polarity and weight for each unit clause. As the weights are given as real values, we prepare a customized version of NuWLS (Chu et al., 2023), that allows the weights to be given as is. During experiments, we run the unchanged solver in parallel with our customized version, and return the result whenever either solver finds a satisfying assignment first.

## 5 EXPERIMENTS

To evaluate the effectiveness of our proposed model, we conduct extensive experiments on various SAT benchmarks, including both crafted and random instances. We compare GranSAT against

Table 1: Architectural features of GranSAT, NeuroSAT, and GAT-SAT. V: Variables, L: Literals, C: Clauses.

| Model | Graph type | Edge features | Logic awareness | Recurrent memory | Attention |
|---|---|---|---|---|---|
| GranSAT | V+C | ✓ | ✓ | ✓ | ✓ |
| NeuroSAT | L+C | ✗ | ✗ | ✓ | ✗ |
| GAT-SAT | L+C | ✗ | ✗ | ✓ | ✓ |

Table 2: Solving Accuracy of models trained on the same distribution as the test set. Rows grouped by difficulty: Easy (top), Hard (bottom). Mean ± std. over 3 runs. Best results in bold.

| Model | CA | SR | 3-SAT | PS | $k$-Clique | $k$-DomSet | $k$-VerCov |
|---|---|---|---|---|---|---|---|
| GranSAT | **67.5 ± 3.6** | **67.1 ± 2.3** | **73.7 ± 1.1** | **74.0 ± 1.1** | 68.3 ± 3.2 | 81.4 ± 4.8 | **86.6 ± 3.0** |
| NeuroSAT | 61.2 ± 2.9 | 59.8 ± 0.9 | 60.2 ± 1.5 | 61.2 ± 1.3 | 70.2 ± 3.7 | 60.6 ± 52.5 | 54.3 ± 47.1 |
| GATSAT | 64.0 ± 2.2 | 61.2 ± 2.3 | 59.7 ± 3.2 | 68.4 ± 1.6 | 70.1 ± 2.0 | 27.6 ± 47.7 | 0.0 ± 0.0 |
| GranSAT | 22.8 ± 1.8 | **12.0 ± 1.0** | **18.2 ± 2.1** | **38.9 ± 3.6** | **29.4 ± 1.4** | 75.7 ± 2.1 | **72.1 ± 3.7** |
| NeuroSAT | 22.8 ± 3.3 | 8.6 ± 2.2 | 9.4 ± 1.0 | 28.9 ± 0.7 | 1.1 ± 0.3 | 23.9 ± 41.5 | 50.7 ± 3.2 |
| GATSAT | **25.0 ± 2.6** | 9.1 ± 2.7 | 11.4 ± 2.2 | 28.0 ± 1.5 | 0.0 ± 0.0 | 0.0 ± 0.0 | 0.0 ± 0.0 |

existing neural SAT solvers, as well as state-of-the-art discrete and continuous SAT solvers. We specifically use GranSAT with a hidden/logical state dimension of 32, attention head number of 4 and the Gödel t-conorm for clause satisfaction, as it showed the best performance in our preliminary experiments.

## 5.1 TRAINING COMPARISON

First, we evaluate the performance of GranSAT against existing neural SAT solvers on various benchmarks provided in the G4SATBench (Li et al., 2024). The benchmarks consist of both crafted and random instances, with varying levels of difficulty. The specific datasets and distributions used are summarized in Appendix C.

As a comparison, we use the following neural models[1]:

- **NeuroSAT** (Selsam et al., 2019): A RNN based model for SAT solving.
- **GAT-SAT** (Chang et al., 2022): A RNN based model with GAT layers applied before the readout phase. [2]

To suit the task of assignment prediction, we specifically modified the models to be trained using the same unsupervised loss function as GranSAT (Equation 2), to ensure a fair comparison. The key architectural features of each model are summarized in Table 1.

**Hyperparameter tuning.** For each dataset subset (consisting of 1000 training, 100 validation, and 1000 test instances), we performed hyperparameter optimization with Optuna (Akiba et al., 2019). We ran 3 independent trials, each training the model for 100 epochs on the training set. The best configuration was selected according to validation performance. We then retrained this configuration on the same training set for a total of 1000 epochs, and report the final results on the held-out 1000 test instances over 3 independent runs. The specific architectures and settings used are detailed in Appendix B.

**Results.** Table 2 summarizes the solving accuracy of each model on various datasets, grouped by difficulty level. The results show that GranSAT outperforms both NeuroSAT and GAT-SAT across all but two datasets (k-Clique on easy instances, and CA on hard instances), demonstrating

---

[1]We implement models using the original papers' descriptions.

[2]We could not use SAT-GATv2 (Chang & Liu, 2025), a successor of GAT-SAT, as implementation details were not made available.

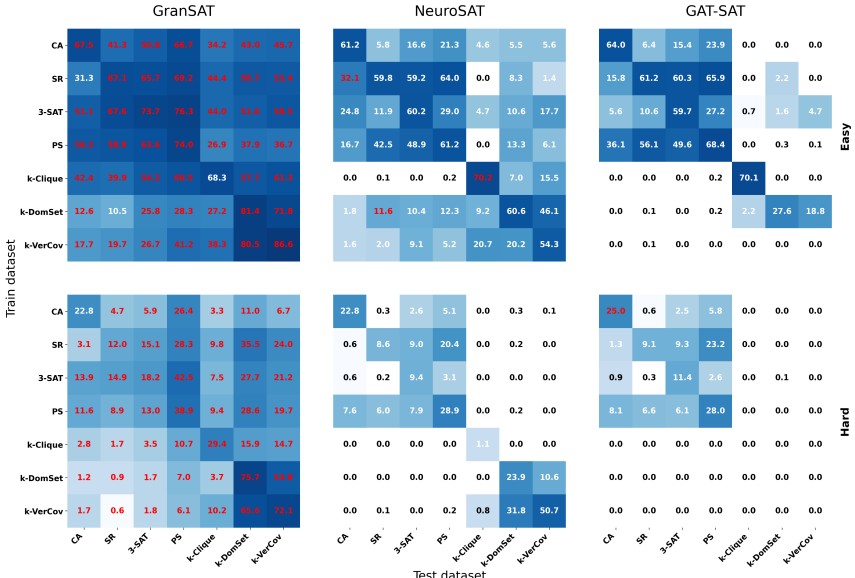

Figure 4: Heatmap of solving accuracy across different training and testing datasets. Figures on top are for easy instances, while the bottom ones are for hard instances. The best results for each train–test pairs is highlighted in red.

Table 3: Solving accuracy of models trained on easy instances, when evaluated on medium instances of the same family. Mean ± std. over 3 runs. Best results in bold.

| Model | CA | SR | 3-SAT | PS | $k$-**Clique** | $k$-**DomSet** | $k$-**VerCov** |
|---|---|---|---|---|---|---|---|
| GranSAT | **15.6 ± 0.3** | **8.2 ± 0.8** | **15.5 ± 1.1** | **22.4 ± 2.1** | **41.1 ± 6.4** | **44.7 ± 3.6** | **33.7 ± 2.3** |
| NeuroSAT | 7.7 ± 0.9 | 5.9 ± 0.4 | 8.1 ± 0.5 | 21.0 ± 4.5 | 17.0 ± 12.1 | 13.7 ± 10.1 | 14.0 ± 1.1 |
| GAT-SAT | 9.3 ± 1.4 | 7.0 ± 0.3 | 7.6 ± 0.7 | 20.5 ± 0.7 | 10.3 ± 14.6 | 0.0 ± 0.0 | 21.0 ± 0.2 |

its effectiveness in learning to solve specific families of SAT problems. The variance across runs are also stable and low, indicating that GranSAT is robust to random initialization and training data sampling. Furthermore, GranSAT retains significantly higher performance on hard instances compared to the other models, indicating that it is able to learn from more complex structures and patterns in the data.

Figure 4 shows a heatmap of solving accuracy across different testing and training datasets. The results indicate that the generalization performance of GranSAT is far superior to that of NeuroSAT and GAT-SAT, as it achieves the best performance (highlighted in red) in all but 3 out of 49 train–test pairs for easy instances, and 1 out of 49 pairs for hard instances, as well as huge performance improvements of up to $60\%$ in some cases. It is also notable that GranSAT trained on random 3-SAT instances shows strong performance across a wide range of datasets, even surpassing models trained on the same distribution in some cases. Overall, the results suggest that GranSAT is able to learn to solve SAT problems in a more generalizable manner compared to existing neural SAT solvers.

Table 3 shows the results of GranSAT trained on easy instances and tested on medium ones from the same family. The results show that GranSAT can generalize to larger instances, reaching non-trivial accuracy across all datasets. This suggests that GranSAT learns useful patterns and structures that go beyond the training data, allowing it to handle bigger and more complex problems, a crucial aspect for practical applications. Notably, GranSAT performs exceptionally well on $k$-DomSet and $k$-VerCov, suggesting that it is particularly effective for structured problems.

One discussion to be had is the size of the training dataset. While we use 1000 training instances for each dataset, others tend to use millions of instances (Selsam et al., 2019; Chang et al., 2022). We argue that the strong performance of GranSAT even with a small training set, is due to its architecture

Table 4: Mean PAR-2 score and number of timeouts (out of total instances) on the benchmark instances from the random track of the SAT Competition 2018.

| Method | PAR-2 | timeouts | Method | PAR-2 | timeouts |
|---|---|---|---|---|---|
| NuWLS | 590.826 | 53/195 | MatSat | 1687.22 | 162/195 |
| GranSAT-NuWLS[+] | **575.366** | **52/195** | GranSAT-MatSat[+] | 1622.29 | 157/195 |
| GAT-SAT-NuWLS[+] | 614.866 | 57/195 | GAT-SAT-MatSat[+] | 1640.64 | 158/195 |
| NeuroSAT-NuWLS[+] | 588.917 | 54/195 | NeuroSAT-MatSat[+] | 1594.85 | 153/195 |
| NSNet-Sparrow | 585.994 | 55/195 | Sparrow2Riss | 598.123 | 57/195 |
| NLocalSAT-probSAT[+] | 784.689 | 75/195 | YalSAT | 1106.98 | 107/195 |

that closely aligns with distributed local search methods, giving it a strong inductive bias towards solving SAT problems.

## 5.2 SAT WITH LOCAL SEARCH POST-PROCESSING

To evaluate the performance of GranSAT in comparison to state-of-the-art discrete and continuous SAT solvers, we conduct experiments on the random track of SAT Competition 2018 (Heule et al., 2019) (the most recent random track), which consists of 255 satisfiable instances with a timeout of 1000 seconds. However, due to GPU memory limits, we only evaluate on instances with file sizes below 64MB, resulting in a total of 195 instances. Specifics of the dataset are provided in Appendix C.2.

We specifically use each of the models trained on the easy 3-SAT dataset, as a majority of the instances in the SAT Competition 2018 random track are randomly generated 3-SAT instances. We denote the solvers that use NuWLS and MatSat as post-processors as NuWLS[+] and MatSat[+], respectively. Additionally, we provide two state-of-the-art local search solvers, Sparrow2Riss (Heule et al., 2018) (winner of 2018 Random Track) and YalSAT (Biere, 2014), as baselines. Furthermore, we compare with two neural-based solvers that combine GNNs with local search methods, NLocalSAT Zhang et al. (2020) and NSNet Li & Si (2022). The best performing combination was chosen for NLocalSAT.

**Results.** Table 4 reports mean PAR-2 scores and timeouts for the SAT Competition 2018 random track. With NuWLS, GranSAT-NuWLS[+] achieves the best score (573.8) with the lowest number of timeouts, outperforming GAT-SAT, NeuroSAT, and Sparrow2Riss, the winner of the track. This shows that the design of GranSAT enables it to capture useful structure in SAT problems, and its outputs further improve performance when combined with a strong solver like NuWLS. In the MatSat setting, GranSAT-MatSat[+] is competitive but slightly behind NeuroSAT-MatSat[+]. We believe this is because GranSAT's outputs are already close to local optima of an objective similar to MatSat's, leaving less room for further improvement. In contrast, NeuroSAT may give weaker and less aligned outputs, which MatSat can adjust more easily. Still, GranSAT-MatSat[+] performs better than plain MatSat, showing that our neural module is capturing useful information.

## 6 CONCLUSION

In this paper, we introduced GranSAT, a novel GNN architecture for SAT solving that operates on fuzzy logic variable assignments. GranSAT employs a two-step procedure, evaluating clause satisfaction using t-conorms and updating assignments with attention mechanisms, closely aligning with distributed local search methods. Furthermore, this is done with hidden states that retain information from previous iterations, allowing the model to make more informed decisions during the update process. Our experimental results demonstrate that GranSAT outperforms existing neural SAT solvers in both performance and generalization across various benchmarks. Furthermore, when combined with local search post-processors, GranSAT achieves state-of-the-art performance on random instances, showcasing its effectiveness in solving SAT problems. Future work includes exploring more advanced attention mechanisms and integrating GranSAT with other types of solvers to further enhance its capabilities.

REPRODUCIBILITY STATEMENT

We provide an anonymized repository containing the full implementation of GranSAT, data pre-processing scripts, and training configuration files. The repository also includes instructions for dataset generation and links to the benchmark datasets (G4SATBench and SAT Competition 2018). All hyperparameter ranges, final chosen values, and training schedules are detailed in Appendix B. Experiments were run on specified hardware (Appendix B), but our implementation is hardware-agnostic and can be reproduced on any machine with sufficient GPU memory. Random seeds are fixed by default to ensure consistent results across runs. Together, these resources enable extremely high reproducibility of our experiments.

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

# A  MODEL SPECIFICS

## A.1  GRU HIDDEN-STATE UPDATE WITH LAYERNORM

The hidden state for each node $h_i^H$ is updated using a Layer-Normalized GRU. Let $h_i^L$ be the current logical state and $h_i^H$ the previous hidden state. We apply feature-wise LayerNorm (with learnable affine parameters) to the pre-activations of the gates and candidate state:

$$\mathbf{z}_i = \sigma\big(\mathrm{LN}\big(\mathbf{W}_z \boldsymbol{h}_i^L + \mathbf{U}_z \boldsymbol{h}_i^H + \mathbf{b}_z\big)\big),$$
$$\mathbf{r}_i = \sigma\big(\mathrm{LN}\big(\mathbf{W}_r \boldsymbol{h}_i^L + \mathbf{U}_r \boldsymbol{h}_i^H + \mathbf{b}_r\big)\big),$$
$$\tilde{\mathbf{h}}_i = \tanh\big(\mathrm{LN}\big(\mathbf{W}_h h_i^L + \mathbf{U}_h (\mathbf{r}_i \odot \boldsymbol{h}_i^H) + \mathbf{b}_h\big)\big),$$
$$h_i^{H'} = (1 - \mathbf{z}_i) \odot \boldsymbol{h}_i^H + \mathbf{z}_i \odot \tilde{\mathbf{h}}_i,$$

where $\sigma$ is the sigmoid function and $\odot$ denotes element-wise multiplication. The same GRU-Norm cell (parameter set) is reused at every iteration/layer, with hidden dimensionality $d$ matching the model specification in the main paper. When LN is the identity map, this reduces to the standard GRU.

## A.2  POLARITY HANDLING

For connections where variables have negative polarity, we pass the negated logical feature, i.e., use $\boldsymbol{x}_i^L \leftarrow 1 - \boldsymbol{x}_i^L$ in the GRU input. Likewise, clause hidden updates use the unsatisfaction signal $(1 - \boldsymbol{c}_j^L)$ as input so that unsatisfied clauses exert stronger influence during subsequent attention and message passing.

## A.3  INPUT/OUTPUT PROCEDURES

**Inputs.** GranSAT initializes each variable node $i$ with a $d$-dimensional logical state $\boldsymbol{x}_i^L \in [0,1]^d$, where each dimension is sampled independently from $\mathrm{Uniform}(0,1)$. Hidden states are set to zero. Clause satisfactions are then computed by applying the chosen t-conorm $T^\vee$ over the literals of each clause.

**Outputs.** After $T$ iterations, each variable node produces a final logical vector $\boldsymbol{x}_i^L$. We obtain the per-variable scalar prediction by averaging across dimensions:

$$p_i = \frac{1}{d} \sum_{k=1}^{d} \big(x_{i,k}^L\big) \in [0,1].$$

Clause satisfactions $\bar{c}_j$ are also obtained by applying the same t-conorm $T^\vee$ over the final literal predictions. These scalar values $p_i$ and $\bar{c}_j$ are used in the loss, evaluation, and post-processing (NuWLS$^+$, MatSat$^+$).

# B  TRAINING SETTINGS

We trained all models using the settings described below. Hyperparameter search and retraining protocols follow the descriptions in the main paper.

## B.1  REPRODUCIBILITY

An anonymized code repository with preprocessing scripts and configuration files is available at `https://anonymous.4open.science/r/GranSAT-ICLR2026`.

## B.2  HARDWARE INFORMATION

Experiments were run on a workstation with the following specifications:

- CPU: AMD Ryzen Threadripper PRO 5995WX (64 cores)

- Memory: 258 GiB RAM
- GPUs: $2 \times$ NVIDIA RTX 6000 Ada Generation (48 GB each), $2 \times$ NVIDIA RTX A6000 (48 GB each)
- CUDA version: 12.6, NVIDIA driver 560.35.05

### B.3 HYPERPARAMETER RANGES

Table 5 summarizes the hyperparameter search space used in our experiments. The learning rate, weight decay, and gradient clipping value were optimized using Optuna (Akiba et al., 2019).

Table 5: Optuna search space used in our experiments.

| Parameter | Domain | Distribution |
|---|---|---|
| Learning rate | $[1 \times 10^{-4},\, 3 \times 10^{-3}]$ | Log-uniform |
| Weight decay | $[1 \times 10^{-6},\, 1 \times 10^{-3}]$ | Log-uniform |
| Gradient clip | $[0.0,\, 2.0]$ | Uniform |

### B.4 ARCHITECTURAL SETTINGS

Table 6 summarizes the iteration count, hidden dimension, number of attention heads, and batch size used for each model in our experiments. Unless otherwise noted, the number of logic-update/message-passing iterations was fixed to 5 for a fair comparison across models.

Table 6: Architectural settings used in our experiments. Iterations denote the number of logic-update/message-passing rounds. Hidden Dim is the per-node hidden width.

| Model | Iterations | Hidden Dim | Att. Heads | Batch Size |
|---|---|---|---|---|
| NeuroSAT | 26 | 128 | - | 32 |
| GAT-SAT | 26 | 128 | 8 | 32 |
| GranSAT | 16 | 32 | 4 | 128 |

## C DATASETS

### C.1 GENERATED DATASET DISTRIBUTION

Tables 7a and 7b provide the value ranges used for generating Easy and Hard benchmarks, respectively. Here, $n$ refers to the number of Boolean variables for problems such as 3-SAT, SR, CA, and PS. For problems such as $k$-Clique, $k$-DomSet, and $k$-VerCov, $v$ denotes the number of vertices in the underlying graph, and $k$ is the size of the subset (e.g., clique size, dominating set size) the problem asks to find. These parameters control both the scale and hardness of the generated instances. It is notable that the Hard dataset was denoted as a Hard dataset in the original G4SATBench (Li et al., 2024), but since only up to the mediums were used for training and evaluation, we refer to it as Hard in this paper.

(a) Description of the Easy benchmark set.

| Problem Family | Easy |
|---|---|
| SR, 3-SAT, CA, PS | $n \in [10, 40]$ |
| $k$-Clique | $v \in [5, 15],\, k \in [3, 4]$ |
| $k$-DomSet | $v \in [5, 15],\, k \in [2, 3]$ |
| $k$-VerCov | $v \in [5, 15],\, k \in [3, 5]$ |

(b) Description of the Hard benchmark set.

| Problem Family | Hard |
|---|---|
| SR, 3-SAT, CA, PS | $n \in [40, 200]$ |
| $k$-Clique | $v \in [15, 20],\, k \in [3, 5]$ |
| $k$-DomSet | $v \in [15, 20],\, k \in [3, 5]$ |
| $k$-VerCov | $v \in [10, 20],\, k \in [6, 8]$ |

Table 7: Variable ranges for the Easy and Hard benchmark sets.

## C.2 SAT COMPETITION 2018 RANDOM TRACK: DATA PROFILE

We summarize the composition of the SAT Competition 2018 Random track instances used in our experiments, split by file size (threshold at 64 MB due to GPU limitations).

Table 8: SATComp2018 Random track composition by file size bucket.

| Bucket | Problems | 3-SAT | Fraction of 3-SAT |
|---|---|---|---|
| Below 64 MB | 195 | 171 | 87.69% |
| Above 64 MB | 60 | 20 | 33.33% |

These statistics justify training/evaluation choices reported in the main text (Sec. 4.6 and Tab. 4). Most sub-64 MB instances are 3-SAT ($\approx 88\%$), making the Easy 3-SAT–trained models a natural fit; larger instances contain a more heterogeneous mix ($\approx 33\%$ 3-SAT).

## D ADDITIONAL EXPERIMENTS

### D.1 GENERALIZATION ON LARGER INSTANCES

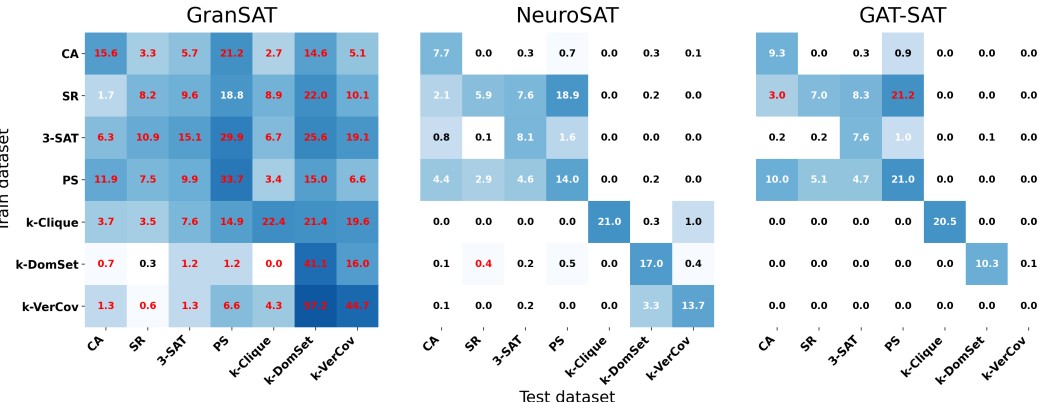

Figure 5: Generalization accuracy: models trained on easy instances, evaluated on medium instances.

Here we show the generalization performance of GranSAT and other models trained on easy instances, and tested on hard instances. The results are summarized in Figure 5. The results show that GranSAT is able to retain strong performance even when the instance size is increased, while the other models show a significant drop in performance. This indicates that GranSAT is able to learn more generalizable features and patterns from the training data, allowing it to perform well on larger and more complex instances.

### D.2 ABLATION: EFFECT OF T-CONORM CHOICE ON EASY 3-SAT

We evaluate how the choice of t-conorm used to compute clause satisfactions impacts learning on the easy 3-SAT distribution. We keep the network architecture and training protocol fixed (hidden/logical dimension $d=32$, 4 attention heads, iteration count as in Table 6) and vary only the t-conorm: Gödel, Łukasiewicz, and Product. Figure 6 reports solving accuracy across training epochs (mean $\pm$ std over 3 independent runs with different seeds).

**Results.** Gödel consistently performs best, converging to ∼0.60–0.65 accuracy by late training. Product is competitive early and stabilizes slightly below Gödel (∼0.53–0.58). Łukasiewicz trails significantly (∼0.10–0.20) throughout. One possible reason for this gap is how each operator works with our update rule: Gödel (a max-type disjunction) gives clear, strong signals that highlight the most useful literals, while Product gives smoother signals but tends to undervalue clauses that are

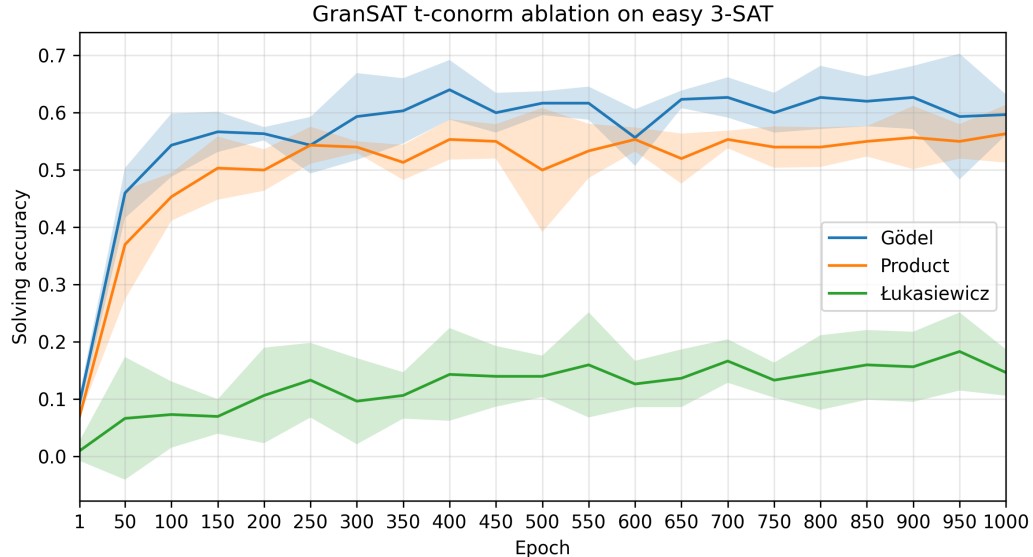

Figure 6: t-conorm ablation on easy 3-SAT Solving accuracy vs. epoch (mean $\pm$ std over 3 runs). Gödel performs best, Product is second, and Łukasiewicz lags due to saturation.

nearly satisfied. Łukasiewicz hits its upper cap too early, which weakens gradients and makes it harder for the model to learn. These patterns appear consistently across different runs.

## E    METRICS AND EVALUATION DETAILS

We report only Solving Accuracy and PAR-2 as our main metrics for evaluation. Unless otherwise stated, values are reported as mean $\pm$ std over 3 independent runs (different random seeds) of the same model configuration.

### E.1    SOLVING ACCURACY (ACC)

The fraction of instances for which the binarized assignment $\hat{x}$ satisfies all clauses:

$$\text{Acc} = \frac{1}{|\mathcal{D}|} \sum_{F \in \mathcal{D}} \mathbf{1}[\hat{x} \text{ satisfies } F].$$

This is the primary metric for the G4SATBench experiments.

### E.2    PAR-2 (PENALIZED AVERAGE RUNTIME)

For a timeout horizon $T = 1000$ seconds, each instance contributes

$$s(F) = \begin{cases} t(F) & \text{if solved within } T, \\ 2T & \text{if not solved within } T, \end{cases}$$

where $t(F)$ is the wall-clock time to produce a satisfying assignment. For hybrid methods (e.g., GranSAT+NuWLS, GranSAT+MatSat), $t(F)$ includes both the downstream solver's runtime and the GNN forward pass plus serialization/export of predictions, ensuring that the total runtime reflects both neural inference and local search.

