# OpenReview forum: "Graph Recurrent Attention Networks for Solving Satisfiability Problems"
_ICLR.cc/2026/Conference — Submitted to ICLR 2026_

### Official Review · Reviewer_7m2n · 2025-10-30

**Soundness:** 2
**Presentation:** 2
**Contribution:** 2
**Rating:** 2
**Confidence:** 4

**Summary:**

This paper propose GranSAT, a GNN-based model for SAT solving. The soft assignments produced by the model can be refined by post-processing with a local search SAT solver. The experimental results show that GranSAT outperforms NeuroSAT and GAT-SAT on G4SATBench and achieves better results on SAT competition 2018 random track than the baseline solvers.

**Strengths:**

1. The unsupervised training objective makes model training more efficient.
2. Combining with classical solvers yields better performance. For example, GranSAT+NuWLS improves PAR-2 and reduces timeouts over strong local-search baselines on the SAT’18 Random track.

**Weaknesses:**

1. Baselines appear dated. The comparisons focus on NeuroSAT and GAT-SAT, which are no longer state of the art among neural SAT solvers. More recent neural and hybrid approaches, SATformer[1], NeuroBack[2] and NLocalSAT[3] are not evaluated. Because these methods also use neural models both to act as neural solver and to guide CDCL or LS solving, comparisons against them should be important.
[1] Satformer: Transformer-based unsat core learning
[2] NeuroBack: Improving CDCL SAT Solving using Graph Neural Networks
[3] NLocalSAT: Boosting Local Search with Solution Prediction

2. The experiment is not convincing enough. Experiments are limited to G4SATBench and the SAT’18 Random track, which are relatively easier. More evaluations should include the latest SAT Competition benchmarks.

**Questions:**

1. Please justify the above weakness.
2. I wonder in Table 2, do all three solvers include post-processing? or are they just used to predict assignments without any LS solvers, like NeuroSAT default settings?

---

> ### Author Response · Authors · 2025-11-28
>
> We thank the reviewer for the constructive feedback. Below we address the concerns raised.
>
> > [W1] Baselines appear dated
>
> We agree that it would be beneficial to compare with more baselines. We have included additional comparisons with NLocalSAT and NSNet in the revised version of the paper (Table 4). However, we did not include SATformer and NeuroBack as both works focuses on integrating with a CDCL-based solver, and neither work studies nor reports performance when integrated with local search. A primary objective of our experiments is to evaluate whether a pipeline of neural SAT solvers with local search post-processing can achieve state-of-the-art performance in random SAT solving.
>
> Below, we provide a table comparing the performance of our combined framework with these baselines on the SAT Competition 2018 Random Track:
>
> | Solver                | PAR-2 Score | Timeouts   |
> |-----------------------|------------:|-----------:|
> | GranSAT+NuWLS         | **575.366** | **52/195** |
> | NSNet+Sparrow         | 585.994     | 55/195     |
> | NLocalSAT+probSAT     | 784.689     | 75/195     |
>
> For a fair comparison, the backbone models are trained on the same dataset and loss function as GranSAT. As the table shows, our combined framework outperforms these baselines, achieving the best PAR-2 score and the least number of timeouts.
>
> > [W2] The experiment is not convincing enough, and the dataset is relatively easier.
>
> In our first experiment, we use G4SATBench instances to specifically evaluate the generalizability of GranSAT and other neural models. As shown in Figure 4, GranSAT consistently outperforms existing neural SAT solvers in terms of generalization across different synthetic datasets, demonstrating the effectiveness of the proposed logic-aware variable updates in comparison to learned message passing updates.
>
> In our second experiment, we evaluate the performance of GranSAT with local search post-processing on the SAT Competition 2018 Random Track, the most recent random track. As shown in Table 4, GranSAT with NuWLS post-processing outperforms state-of-the-art local search solvers including Sparrow2Riss, the best performing solver in the SAT Competition 2018 Random Track, as well as other non-neural local search solvers such as YalSAT and probSAT.
> Furthermore, we note that the competition instances are very challenging; even state-of-the-art SAT solvers fail to solve more than 50 instances within the time limit, as shown in Table 4. Our results demonstrate that GranSAT with local search post-processing can achieve state-of-the-art performance in random SAT solving, even on these challenging benchmarks.
>
>
> > [Q2] In Table 2, do all three solvers include post-processing?
>
> No, the results in Table 2 do not include any post-processing; they are evaluated based on the direct predictions from the models. It shows what percentage of problems the neural models were able to provide completely satisfying assignments for without any further refinement.
>
>
>
> Overall, we would like to reiterate that a major contribution of GranSAT is the *logic-aware* variable update mechanism described in Sections 4.2 and 4.3. Existing neural SAT solvers mainly rely on learned message passing updates for both variables and clauses, with no explicit use of fuzzy-logic operators for evaluating clause satisfaction at each step.
> In contrast:
> 1. GranSAT introduces t-conorm-based clause satisfaction computation together with attention-based variable updates that considers the satisfaction state of clauses—when a clause’s satisfaction is low (respectively high), the assignments are pushed toward increasing (respectively maintaining) it.
> 1. While prior works use the hidden states mainly for prediction, GranSAT uses them as heuristics to guide the logic-aware variable updates. The recurrent design also allows the model to retain historical information, similar to local search methods that use history to avoid cycles or poor local minima.
>
> These design choices allow GranSAT to perform distributed local search with learnable heuristics, making it a non-trivial extension over existing GNN-based SAT solvers. Furthermore, the results in Figure 4 show that GranSAT consistently generalizes better across synthetic datasets compared to existing models with learned updates, demonstrating the effectiveness of the proposed logic-aware updates.

---

### Official Review · Reviewer_ZKSC · 2025-10-31

**Soundness:** 3
**Presentation:** 4
**Contribution:** 3
**Rating:** 4
**Confidence:** 4

**Summary:**

This paper presents GranSAT, a hybrid neural SAT solver based on Graph Recurrent Attention
Networks that operate on fuzzy logic variable assignments. Clause satisfaction is computed deterministically using t-conorm fuzzy operators (Gödel, Łukasiewicz, or Product), while variable assignments are updated via attention-based message passing combined with recurrent hidden states. This continuous relaxation of Boolean logic allows gradient-based learning while maintaining logical structure. When paired with local-search post-processing (NuWLS, MatSat), GranSAT achieves strong empirical performance on several benchmark families (G4SATBench and SAT Competition 2018).

**Strengths:**

• Clear and mathematically rigorous formulation.
• Elegant integration of fuzzy logic with attention and recurrence.
• Strong empirical performance on multiple benchmarks.
• Transparent experimental protocol and reproducibility.

**Weaknesses:**

• Most components are well-known; innovation lies mainly in using fixed fuzzy operators.
• The claimed relation between attention and local search is not supported by analysis.
• Outdated baselines; unclear definition of “state of the art.”
• Related Work section lacks critical comparison.

**Questions:**

• Can the authors justify or empirically demonstrate the claimed connection between attention
and distributed local search?
• Does fixing clause updates via t-conorms lead to measurable advantages over learned
updates?
• How sensitive is performance to the choice of t-conorm?

---

> ### Author Response · Authors · 2025-11-28
>
> We thank the reviewer for insightful comments and suggestions to improve our paper. We have carefully addressed the concerns raised in the review as follows:
>
>
> > [W1] Most components are well-known; innovation lies mainly in using fixed fuzzy operators.
>
> While it is true that the components used in GranSAT, such as attention mechanisms and fuzzy logic operators, are well-known individually, our main contribution lies in the novel integration of these components to create a powerful SAT solver. Specifically, we introduce a logic-aware variable update mechanism that combines t-conorm-based clause satisfaction computation with attention-based updates that consider clause satisfaction states. This design allows GranSAT to perform distributed local search with learnable heuristics, which is a non-trivial extension over existing GNN-based SAT solvers.
>
> > [W2/Q1] The claimed relation between attention and local search is not supported by analysis.
>
> One of the main contributions of our work comes from the attention-based logical updates, as shown in Section 4.2 and 4.3. The updates are done in the following steps, given a fuzzy logical assignment for variables:
> 1. The clause satisfaction is computed based on variable logical assignments using t-conorms.
> 1. The variable decides which clauses to prioritize based on the current hidden state, using attention mechanisms.
> 1. The variable takes all clause satisfactions into account, and updates its logical value to satisfy the clauses deemed important by the attention mechanism.
>
> It is noted that the logical values are not obtained by post-processing the hidden states, but are the model’s internal assignments that directly drive clause satisfaction. Furthermore, the update mechanism closely resembles local search, which is mainly performed using the following steps:
> 1. Evaluate the current assignment and identify unsatisfied clauses.
> 1. Select a variable to flip based on some heuristic on the unsatisfied clauses.
> 1. Flip the variable and repeat.
>
> Moreover, by combining these update mechanisms with RNNs, we can retain the history of previous updates, allowing for more informed decisions in future updates similar to how local search methods may use historical information to avoid cycles or poor local minima.
> These design choices support our claims regarding the relation between fuzzy logic/attention-based updates and distributed local search.
>
>
>
> > [W3] Outdated baselines and unclear definition of “state of the art.”
>
> We thank the reviewer for this comment. We have included additional comparisons with works that focus on random SAT solving, namely NLocalSAT and NSNet, in the revised version of the paper.
>
> | Solver                | PAR-2 Score | Timeouts   |
> |-----------------------|------------:|-----------:|
> | GranSAT+NuWLS         | **575.366** | **52/195** |
> | NSNet+Sparrow         | 585.994     | 55/195     |
> | NLocalSAT+probSAT     | 784.689     | 75/195     |
>
> For a fair comparison, the backbone models are trained on the same dataset and loss function as GranSAT. As the table shows, our combined framework outperforms these baselines, achieving the best PAR-2 score and the least number of timeouts.
>
> We have compared to state-of-the-art local search solvers including Sparrow2Riss, the best performing solver in the SAT Competition 2018 Random Track, as well as other non-neural local search solvers such as YalSAT and probSAT.
> As shown in Table 4, GranSAT with NuWLS post-processing outperforms these solvers, achieving state-of-the-art performance in random SAT solving.
>
> > [W4] Related Work section lacks critical comparison.
>
> In Section 2.1, we have a paragraph dedicated to providing the differences between GranSAT and existing neural SAT solvers, with the two main differences being:
> 1. T-conorm based clause evaluation and attention based variable updates that consider clause satisfaction states.
> 1. Use of continuous and discrete local search methods as post-processing to obtain satisfying assignments.
>
> Furthermore, we provide Section 2.2 as a reference for readers to understand what the existing local search solvers are, and not as a direct comparison with our method, as these solvers do not incorporate neural networks.

---

> > ### Author Response · Authors · 2025-11-28
> >
> > > [Q2] Does fixing clause updates via t-conorms lead to measurable advantages over learned updates?
> >
> > While it is possible to introduce learned mappings for clause value computation, our goal is to ensure that the value consistently represents clause satisfaction.
> > This property is essential for the attention-based updates, which use the clause value to determine how the variable assignments should be adjusted—when a clause’s satisfaction is low (respectively high), the assignments are pushed toward increasing (respectively maintaining) it.
> > Furthermore, in comparison to existing models such as NeuroSAT and GAT-SAT that utilize learned message passing updates for both variables and clauses, we can observe that GranSAT consistently outperforms them in terms of performance across datasets, as shown in Figure 4. This demonstrates the effectiveness of using t-conorms along with attention-based logical updates for variable assignments.
> >
> >
> > > [Q3] How sensitive is performance to the choice of t-conorm?
> >
> > We have provided an ablation study on the choice of t-conorm in Appendix D.2. The results show that the choice of t-conorm does affect the performance, with the Gödel t-conorm performing best. We have further observed that Łukasiewicz t-conorm does not perform well, likely due to its function producing 0 for many inputs, leading to vanishing gradients during training.

---

### Official Review · Reviewer_DgoC · 2025-11-01

**Soundness:** 3
**Presentation:** 3
**Contribution:** 3
**Rating:** 6
**Confidence:** 3

**Summary:**

The paper introduces a method for solving SAT problems using a Graph Recurrent Attention Network (GranSAT).
The method:
1.	Evaluates each clause’s satisfaction state under fuzzy variable assignments using t-conorms.
2.	Updates the variable assignments via attention mechanisms.
GranSAT produces fuzzy assignments, which can be refined through local search–based post-processing methods.
The authors conduct experiments on various benchmarks and compare the performance of GranSAT with state-of-the-art neural and discrete SAT solvers.

**Strengths:**

The paper is well written and easy to read.

The authors claim that
•	GranSAT outperforms existing neural SAT solvers
•	When combined with local search post-processors, GranSAT achieves state-of-the-art performance on random SAT instances.

**Weaknesses:**

Due to GPU memory limitations, only 195 out of 255 instances were evaluated.

While SAT problems are interesting, they are quite abstract. In my opinion, it would have been valuable to include some high-impact, real-world applications that can be formulated as SAT problems and solved using GranSAT.

**Questions:**

•	Line 194: Instead of using a t-conorm, could this mapping be replaced with a simple neural network whose weights are trained jointly with the Graph Neural Network?

•	Line 233: In many applications, Transformer- or Mamba-based architectures outperform GRU and LSTM models. Could these architectures be tried here, replacing the GRUs?

•	How sensitive is GranSAT to hyperparameters? I understand that in the experiments, the hidden/logical state dimension was set to 32 and the number of attention heads to 4. Were these values found to be optimal? How much do the results vary with different parameter choices?

•	It is interesting that in the experiments, GAT-SAT performs best on the hard-CA dataset but completely fails on some of the other tasks.

•	What were the wall-clock training times for the experiments?

•	Based on Appendix B, my understanding is that the authors used four GPUs. How were the training tasks distributed across these GPUs?

---

> ### Author Response · Authors · 2025-11-22
>
> We thank the reviewer for insightful comments and suggestions to improve our paper. We have carefully addressed the concerns raised in the review as follows:
>
> > [W] Missing real-world application examples.
>
> As our current framework performs local-search based post-processing on the neural predictions, we focus on random SAT instances, which are the standard benchmarks where stochastic local search methods are typically evaluated.
> Nevertheless, extending GranSAT to structured, real-world SAT problems is an interesting direction that we intend to explore in future work.
>
>
> > [Q1] Instead of using a t-conorm, could this mapping be replaced with a simple neural network whose weights are trained jointly with the Graph Neural Network?
>
> While it is possible to introduce learned mappings for clause value computation, our goal is to ensure that the value consistently represents clause satisfaction.
> This property is essential for the attention-based updates, which uses the clause value to determine how the variable assignments should be adjusted—when a clause’s satisfaction is low (respectively high), the variable assignments are pushed towards increasing (respectively maintaining) the satisfaction. The usage of t-conorms provides the desired behavior by design while keeping the mappings relatively simple.
>
> > [Q2] Could Transformer- or Mamba-based architectures be tried here, replacing the GRUs?
>
> Yes, this is possible. We have chosen the GRUs due to their simplicity and low computational overhead, but we agree that more complex architectures such as Transformers or Mamba-based architectures could potentially improve the performance further. We think this is an interesting direction for future work.
>
> > [Q3] How sensitive is GranSAT to hyperparameters?
>
> From the grid search we have performed, the hidden/logical state dimension of 32 and the number of attention heads of 4 were found to be optimal in our experiments. However, varying these parameters did not lead to significant changes in performance, indicating that GranSAT is relatively robust to hyperparameter choices.
>
> > [Q4] GAT-SAT performs best on the hard-CA dataset but completely fails on some of the other tasks.
>
> We agree that this is an interesting phenomenon. We have also observed from the results in Figure 4 that certain architectures yield higher performance on specific types of datasets. We believe this illustrates how different architectures specialize in different types of problems, whereas GranSAT remains stable across them.
>
>
> > [Q5] What were the wall-clock training times for the experiments?
>
> As the wall-clock training times vary depending on the dataset and model, we provide the training times for each model on the Easy 3-SAT dataset as a reference:
>
> | Model    | Seconds |
> |----------|---------|
> | NeuroSAT | 2555    |
> | GAT-SAT  | 6841    |
> | GranSAT  | 1766    |
>
> This shows that our model is not only more effective but also more efficient in training compared to the baselines.
>
> > [Q6] How were the training tasks distributed across these GPUs?
>
> Each model was trained on a single GPU; the multiple GPUs were used only to run different training jobs in parallel.

---

### Official Review · Reviewer_GJ9i · 2025-11-04

**Soundness:** 2
**Presentation:** 3
**Contribution:** 1
**Rating:** 2
**Confidence:** 5

**Summary:**

This paper proposes, GranSAT, a graph neural network-based approach to solving the classic satisfiability problems. GranSAT uses the graph recurrent attention networks and alternates two updates: 1) clause's satisfaction state update using t-conorms; 2) variable assignment update using attention mechanisms. GranSAT can be used as a standalone solver and integrated with local search methods for further post-processing. Experimental evaluations are performed on the standard G4SATBench, which was designed to evaluate GNNs on sat solving, and GranSAT outperforms two baselines, i.e., NeuroSAT and GAT-SAT. When integrated with local search (NuWLS), GranSAT is close to or slightly outperforms state-of-the-art local search solvers such as Sparrow2Riss and YalSAT.

**Strengths:**

- the paper is generally well-written; particularly, visual illustrations are very helpful for conveying the essential idea
- the targeted problem, Boolean satisfiability, is of great importance, and relevant background about graph neural networks and attention mechanism are properly addressed

**Weaknesses:**

- the proposed method is fairly incremental, given that there are numerous attempts of applying graph neural networks for sat solving since the seminal work NeuroSAT (2019).
- the chosen baselines (i.e., NeuroSAT and GAT-SAT) are quite limited, omitting many important baselines such as DG-DAGRNN, NLocalSAT, QuerySAT, Graph-QSAT, NSNet, to name a few.
- the evaluation results of G4SATBench are inconsistent with the original evaluation of NeuroSAT
- problems of SAT Competition 2018 seem to be quite outdated
- the highlighted application for local search is not promising, i.e., the performance is close to the baseline NeuroSAT and there is a significant gap from the state-of-the-art  local search solver

**Questions:**

The accuracy numbers reported in this work are significantly different from the evaluation in the G4SATBench, particularly the results of NeuroSAT. Why is there such a big difference?

Why is SAT competition 2018 used, instead of the recent competitions (SAT Competition 2024)?

Minor writing issues:
- Notations like $e_{ij+n}$ is confusing, which may refer to $e_k$ where $k=i*j+n$ or $e_{i, j+n}$.
- Furthermore, $h_j$ and $h_{j+n}$ are also confusing. What are possible values of $N_i$? Shouldn't $h_{j+n}$ be used when considering $j \in N_i$.

---

> ### Author Response · Authors · 2025-11-28
>
> We thank the reviewer for insightful comments and suggestions to improve our paper. We have carefully addressed the concerns raised in the review as follows:
>
> > [W1] The proposed method is fairly incremental
>
> A major contribution of GranSAT is the *logic-aware* variable update mechanism described in Sections 4.2 and 4.3. Existing neural SAT solvers including NeuroSAT, GAT-SAT, DG-DAGRNN, NSNet, NLocalSAT, QuerySAT and Graph-QSAT rely on learned message passing updates for both variables and clauses, with no explicit use of fuzzy-logic operators for evaluating clause satisfaction at each step.
> In contrast:
> 1. GranSAT introduces t-conorm-based clause satisfaction computation together with attention-based variable updates that considers the satisfaction state of clauses—when a clause’s satisfaction is low (respectively high), the assignments are pushed toward increasing (respectively maintaining) it.
> 1. While prior works use the hidden states mainly for prediction, GranSAT uses them as heuristics to guide the logic-aware variable updates. The recurrent design also allows the model to retain historical information, similar to local search methods that use history to avoid cycles or poor local minima.
>
> These design choices allow GranSAT to perform distributed local search with learnable heuristics, making it a non-trivial extension over existing GNN-based SAT solvers. These differences are also summarized in Table 1 that highlights the differences in the models we compare against. Furthermore, the results in Figure 4 show that GranSAT consistently generalizes better across synthetic datasets compared to existing models with learned updates, demonstrating the effectiveness of the proposed logic-aware updates.
>
>
>
>
> > [W2] The chosen baselines are quite limited
>
> We thank the reviewer for this comment. We have included additional comparisons with works that focus on random SAT solving combined with local search, namely NLocalSAT and NSNet, in the revised version of the paper. We did not include DG-DAGRNN, as their implementation is not publicly available. Furthermore, we did not include QuerySAT and Graph-QSAT as the former focuses on building a stand-alone neural SAT solver, while the latter focuses on integrating into a CDCL-based solver, and neither work studies nor reports performance when integrated with local search. A primary objective of our experiments is to evaluate whether a pipeline of neural SAT solvers with local search post-processing can achieve state-of-the-art performance in random SAT solving.
>
> Below, we provide a table comparing the performance of our combined framework with these baselines on the SAT Competition 2018 Random Track:
>
>
> | Solver                | PAR-2 Score | Timeouts   |
> |-----------------------|------------:|-----------:|
> | GranSAT+NuWLS         | **575.366** | **52/195** |
> | NSNet+Sparrow         | 585.994     | 55/195     |
> | NLocalSAT+probSAT     | 784.689     | 75/195     |
>
> For a fair comparison, the backbone models are trained on the same dataset and loss function as GranSAT. As the table shows, our combined framework outperforms these baselines, achieving the best PAR-2 score and the least number of timeouts.

---

> > ### Author Response · Authors · 2025-11-28
> >
> > > [W3/Q1] The evaluation results of G4SATBench are inconsistent with the original evaluation of NeuroSAT
> >
> > A major reason contributing to the difference is the size of datasets used. Our dataset consists of 1000 training instances (L365), while the G4SATBench experiments were conducted using 80k instances. Our aim is to evaluate the behavior of the models under limited training data, as discussed in L444-448. To keep the experiments fair, all models use the same dataset and follow the hyperparameter settings given in their respective papers.
> >
> >
> > > [W4/Q2] SAT Competition 2018 seems to be quite outdated
> >
> > As our objective was to evaluate the performance of our framework on random SAT instances, we have chosen to use the SAT Competition 2018 Random Track, the most recently held random track. This is also the case in other works focusing on improving local search methods for random SAT solving, such as NLocalSAT.
> >
> > > [W5] The highlighted application for local search is not promising
> >
> > Our application to local search is promising for the following reasons:
> > 1. Neural network based SAT solvers are only possible of providing approximate solutions. To obtain completely satisfying assignments, symbolic methods such as local search are necessary. Our work demonstrates that by using local search as post-processing, we can elevate the performance of neural SAT solvers to be competitive with state-of-the-art local search solvers.
> > 1. Our combined frameworks, especially with NuWLS, shows to be effective in solving random SAT instances, with GranSAT+NuWLS achieving state-of-the-art performance for random SAT instances as shown in Table 4. Furthermore, the table shows that GranSAT+NuWLS solves an average of 13 seconds faster than the second best solver, and a lower number of timeouts, demonstrating its effectiveness. Additionally, NSNet+Sparrow is shown to outperform NeuroSAT+NuWLS, indicating that without a strong backbone model, the performance gain from local search post-processing is limited.
> >
> > > Minor writing issues
> >
> > We thank the reviewer for pointing these out. We have edited $e_{ij+n}$ to $e_{i,j+n}$, and now we use $\mathcal{N}_j$ consistently to denote the neighborhood of node $j$ to avoid ambiguity.

---

### Meta-Review · Area_Chair_YbkV · 2026-01-06

**Summary:**

The paper proposes GranSAT, a neural SAT solver based on Graph Recurrent Attention Networks. The method introduces a mechanism that evaluates clause truth degrees using t-conorm fuzzy logic operators and updates variable assignments via attention mechanisms coupled with recurrent hidden states. The authors position this approach as a continuous relaxation of local search. Experiments are conducted on the G4SATBench synthetic dataset and the SAT Competition 2018 Random Track, with results comparing against NeuroSAT, GAT-SAT, and (in the rebuttal) NLocalSAT and NSNet.

**Reviewer Concerns:**

**Addressed Concerns:**
The authors made efforts to address the lack of baselines pointed out by Reviewers **GJ9i**, **ZKSC**, and **7m2n** by including comparisons with NLocalSAT and NSNet in the rebuttal. They also provided clarifications regarding the hyperparameter sensitivity and training times in response to Reviewer **DgoC**.

**Outstanding Concerns:**
*   **Novelty and Incrementality:** Reviewers **GJ9i** and **ZKSC** remain concerned that the proposed method is an incremental combination of known components (GNNs, attention, fuzzy logic) without sufficient technical innovation to distinguish it significantly from existing neural SAT solvers.
*   **Theoretical Grounding:** Reviewer **ZKSC** noted that the claimed connection between the attention mechanism and distributed local search is asserted descriptively rather than supported by rigorous analytical evidence.
*   **Benchmark Relevance:** Reviewers **GJ9i** and **7m2n** criticized the reliance on older benchmarks (SAT Competition 2018) and the inconsistency of the reported baseline performance on G4SATBench compared to original literature.
*   **Real-world Applicability:** Reviewer **DgoC** highlighted the lack of evaluation on real-world application instances, as the experiments heavily favor random tracks where the proposed method's advantage is marginal.

**Reviewer Scores:**

**Reviewer GJ9i: 2**
The reviewer would likely maintain their score as the rebuttal did not convincingly resolve the concerns regarding the incremental nature of the work and the inconsistencies in the G4SATBench baseline replications.

**Reviewer DgoC: 6**
The reviewer would likely maintain their score as the authors adequately answered the specific questions about training resources and hyperparameters, though the reviewer noted they would not object to a rejection.

**Reviewer ZKSC: 4**
The reviewer would likely maintain their score because, while new baselines were added, the fundamental concern regarding the theoretical justification for the attention-local search connection remains unaddressed.

**Reviewer 7m2n: 2**
The reviewer would likely maintain their score as the scope of the experiments remains limited to random tracks and older competitions, failing to demonstrate state-of-the-art performance against modern hybrid solvers on broader benchmarks.

---

### Decision · Program_Chairs · 2026-01-26

Reject